# Dual-layer optical encryption fluorescent polymer waveguide chip based on optical pulse-code modulation technique

Chunxue Wang[1], Daming Zhang[1], Jian Yue[1], Xucheng Zhang[1], Hang Lin[1], Xiangyi Sun[1], Anqi Cui[1], Tong Zhang[1], Changming Chen ®[1] ✉ & Teng Fei ®[1] ✉

Information encryption technique has broad applications in individual privacy, military confidentiality, and national security, but traditional electronic encryption approaches are increasingly unable to satisfy the demands of strong safety and large bandwidth of high-speed data transmission over network. Optical encryption technology could be more flexible and effective in parallel programming and multiple degree-of-freedom data transmitting application. Here, we show a dual-layer optical encryption fluorescent polymer waveguide chip based on optical pulse-code modulation technique. Fluorescent oligomers were doped into epoxy cross-linking SU-8 polymer as a gain medium. Through modifying both the external pumping wavelength and operating frequency of the pulse-code modulation, the sender could ensure the transmission of vital information is secure. If the plaintext transmission is eavesdropped, the external pumping light will be switched, and the receiver will get warning commands of ciphertext information in the standby network. This technique is suitable for high-integration and high-scalability optical information encryption communications.

With the rapid development of optical communication technology, the information exchange in daily life has entered the form of high-speed transmission[1-3]. It has become increasingly convenient to communicate and transmit digital information between people through optical networks[4,5]. Fast and efficient information transmission method plays an essential role in the development of the country's science and technology, education, economy, and other fields[6-8]. Especially with the advent of smartphones and WIFI, people can access the Internet to obtain images and information anytime and anywhere[9]. However, much of the information transmitted over the Internet is private that needs to be transmitted between the sender and the receiver, such as privacy, trade secrets, business programs, military operational maps, etc.[10,11]. Other information related to national security, so the encryption protection of these data is increasingly important to all sectors of society.

Since the amount of encrypted data is growing, traditional encryption techniques are increasingly unable to satisfy the demands of high security and large data quantities[12]. In recent years, researchers have developed a generation of optical information security solutions by combining optical code modulation techniques with traditional encryption techniques[13-15]. Optical code modulation encryption techniques have features such as large information content, parallel programming capabilities, and multiple degree-of-freedom[16-18]. Firstly, optical signals have the natural ability to process two-dimensional (2D) information in parallel, which is especially suitable for processing data information with high speed[19]. The more complex the data to be processed and the larger the amount of information, the more prominent this advantage becomes. Secondly, compared with digital encryption systems, optical code modulation encryption methods can be extended to three-dimensions (3D) by integrating various degree-of-

---

[1]State Key Laboratory of Integrated Optoelectronics, College of Electronic Science and Engineering, Jilin University, Changchun 130012, PR China.
✉e-mail: chencm@jlu.edu.cn; feiteng@jlu.edu.cn

freedom of light beams during encryption, such as in wavelength, amplitude, phase, and frequency modulation[20–22]. All these aspects establish a huge key space, which allows optical encryption systems with high security.

Among the variety of optical encryption devices, the structures of functionally integrated polymer optical waveguide devices exhibit the characteristics of structural diversity[23]. From the one-dimensional photonics devices to multi-dimensional photonics devices, optical information transmission can become increasingly integrated by combining various structural optical devices, such as optical routing, optical coupler, and optical beam splitters[24–26]. And in order to enhance the safety of information communication systems, various optical materials are applied according to different functional designs[27,28]. The usage of polymer materials for optical waveguide device fabrication has the following advantages: (i) the fabrication process is simple, and the devices can usually be fabricated in bulk by lithography and etching methods, which could improve the production efficiency and save expenses[29], (ii) the refractive index of polymer materials can be flexibly adjusted to satisfy different needs, and polymers can be easily doped with other organic or inorganic materials[30,31], (iii) there is no restriction on the substrate, and both inorganic crystals and polymer films can be applied as substrates for the waveguide devices[32,33], (iv) the cross-linking structure of polymer materials improves their mechanical strength, and related techniques are mature enough to facilitate the fabrication of integrated photonics encryption chips[34,35].

In this work, dual-layer optical encryption fluorescent polymer waveguide chip based on optical pulse-code modulation technique is proposed to optical encryption communication application. The photo-luminescence (PL) properties of green (TCBzC) and red (TCNzC) fluorescent light-emitting materials were analyzed. TCBzC and TCNzC were doped into the epoxy SU-8 polymer to form encryption waveguide materials with gain properties, respectively. The dual-layer waveguide structure for on-chip pulse-code modulation was optimally designed, and dual-layer waveguide devices can be fabricated by the UV direct writing process method. The luminescence and gain properties of dual-layer encryption waveguide devices were investigated. Subsequently, a demonstration of 405 nm pumping light pulse-code modulation for the dual-layer optical waveguide chip was given, which can transmit data information well when the signal is eavesdropped. The maximum relative gains of TCBzC/SU-8 and TCNzC/SU-8 waveguide at 532 and 655 nm wavelength are 5.71 and 5.34 dB, respectively. The response time could measure to be 260 μs. Following this, the code encryption transmission and warning functions of the waveguide device with pulse-code modulation 532 nm wavelength pumping light was measured. The maximum relative gain at 655 nm wavelength was 7.45 dB, and the response time was measured to be 264 μs. This technique can utilize a 3D polymer photonic integration platform to enable optical pulse-code modulation encryption transmission and warn the leakage of data information. With potential applications in optical pattern display and information encryption, it is expected to be valuable in the profound integration Internet information technique with safety and stability.

## Results

The molecular structure formulas of synthesized fluorescent small molecule oligomers green-light TCBzC and red-light TCNzC are illustrated in Fig. 1a[36]. This structure has several advantages. Firstly, flexible alkyl chains suspended on the main chain endow the oligomers solution processing properties. Secondly, the materials are amorphous, they do not crystallize or aggregate in the film. Meanwhile, different luminescent rigid cores (2,1,3-benzothiadiazole for TCBzC and 2,1,3-naphthalenediazole for TCNzC) are existing in the molecules, so that green and red-light emission could be achieved, respectively. In the experiment, we doped TCBzC and TCNzC with the mass fraction of

5 wt‰ in the epoxy cross-linking SU-8 polymer, respectively, to form waveguide core layer materials with better PL and gain properties. As shown in Fig. 1a, both TCBzC/SU-8 and TCNzC/SU-8 can form transparent and uniform solutions. SU-8 is an epoxy-resin type negative photoresist, and the molecular structure of SU-8 is given in Fig. 1b. The polymer material has a high exposure uniformity, and the waveguide obtained by the photolithography process has a good steepness. In addition, it has fine self-leveling ability, so that well-shaped optical waveguide patterns can be achieved. For the cladding layer material of the polymer waveguide device, we chose methyl methacrylate-propyl methacrylate epoxy copolymer (P(MMA-co-GMA)), which is an epoxy cross-linking acrylic resin material. We can synthesize it in the laboratory with methyl methacrylate (MMA) and propylene oxide methacrylate (GMA) by copolymerization, as shown in Fig. 1b. The refractive index of the self-synthesized P(MMA-co-GMA) film is 1.490 measured by an ellipsometer (SPEL M-2000VI, America) from 500 to 700 nm wavelength, which has a minor birefringence and can be adjusted in a wide range. The varied curves for refractive index ($n$) and extinction coefficient ($k$) in Vis-NIR wavelength region for TCBzC/SU-8 and TCNzC/SU-8 waveguide materials are measured by the ellipsometer. The refractive index of bottom layer waveguide material TCBzC/SU-8 is 1.597 at 532 nm wavelength and for top layer waveguide material TCNzC/SU-8 is 1.587 at 655 nm wavelength as given in Fig. 1c. According to the complex refractive index ($\tilde{n}$) function of $\tilde{n} = n + ik$, $k$ refers to the imaginary part of the optical constant as optical absorption-loss coefficient. Therefore, the value of $k$ could be defined as the main optical-loss coefficient for TCBzC/SU-8 and TCNzC/SU-8 waveguide materials. In Fig. 1c, it could be found that the values of $k$ for both TCBzC/SU-8 waveguide material at 532 nm wavelength and TCNzC/SU-8 waveguide material at 655 nm wavelength are $<3 \times 10^{-4}$ cm$^{-1}$. It is demonstrated that TCBzC/SU-8 waveguide material at 532 nm wavelength and TCNzC/SU-8 waveguide material at 655 nm wavelength have low optical-loss characterization. Moreover, multi-layer waveguide devices can be easily realized by spin-coating technique, which can meet our demand for 3D integrated photonic chip.

### Design and simulation of dual-layer optical encryption waveguide chip

The normalized PL intensity spectra of TCBzC and TCNzC are depicted in Fig. 2a. Emission peaks as 540 and 611 nm wavelength are pumped with the 365 nm wavelength light. Absorption peaks at 430 and 505 nm wavelength in visible light region are obtained[37]. Based on the absorbance spectra, there should be a fluorescence energy resonance transfer (FERT) between TCBzC and TCNzC. In experiment, the dual-layer optical encryption fluorescent polymer waveguide structure is designed and fabricated. TCBzC and TCNzC are solely used as gain medium for top and bottom waveguide layer, respectively. The P(MMA-co-GMA) buffer layer between the top (TCNzC/SU-8) and bottom (TCBzC/SU-8) waveguide layers could effectively avoid the FERT phenomenon and guarantee the performances of the waveguide chip. Compared to 365 nm ultraviolet (UV) wavelength light chosen to analyze PL characteristic of the dye oligomers (TCBzC and TCNzC), 405 nm visible wavelength light is used to pump the actual waveguide chip. The main reason is that 365 nm UV wavelength light might cause photo-bleaching phenomenon for the dye oligomers in polymer waveguide. Contrast to 365 nm wavelength UV light, 405 nm visible wavelength has enough photon energy while hardly result in damage to the dye oligomers in polymer waveguide. Therefore, 405 nm visible wavelength light is used to pump the waveguide chip by optical pulse-code modulation technique in actual experiment. Accordingly, for the waveguide device fabricated by TCBzC/SU-8, with the 532 nm wavelength light as the signal light, the intensity of the output 532 nm signal light will be gained when the enhancement generated in the optical power intensity of the 405 nm pumping light. Depending on the 655 nm wavelength light as key signal source in visible light fiber

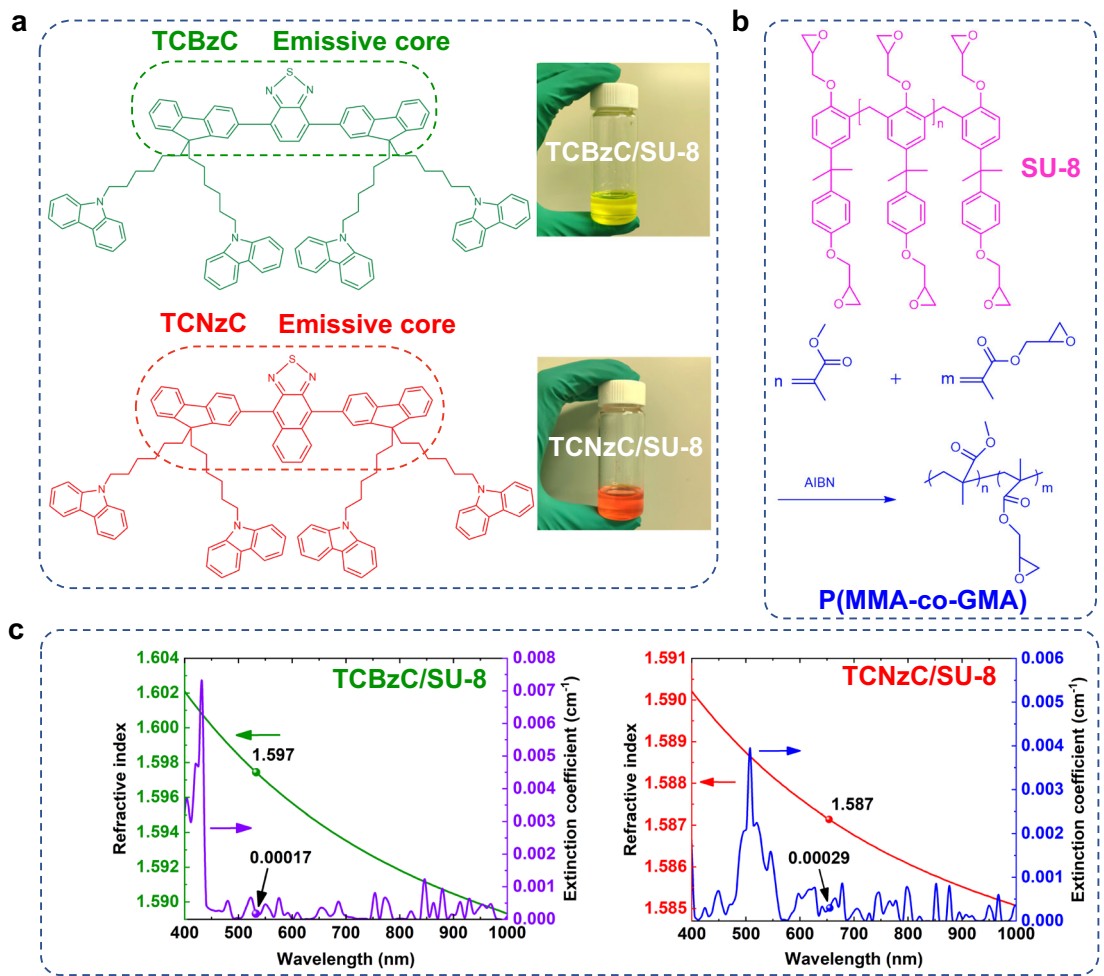

**Fig. 1 | Materials applied to the dual-layer optical encryption fluorescent polymer waveguide chip. a** Molecular structures of TCBzC and TCNzC, with the clarified TCBzC/SU-8 and TCNzC/SU-8 solutions; **b** the molecular structure of SU-8 core layer and the synthesis process for P(MMA-co-GMA); **c** the refractive index (*n*) and extinction coefficient (*k*) characterization of TCBzC/SU-8 and TCNzC/SU-8 optical waveguide materials.

communication system[38,39], it might not be the maximum fluorescence emission peak of TCNzC (at 610 nm length), but there will be greatly potential application in actual optical information transmission network. When the signal light at 655 nm wavelength is coupled into the TCNzC/SU-8 waveguide with 405 or 532 nm pumping light as the external excitation source, the intensity of the signal light output from the waveguide device will also be enhanced. Based on the above, the 3D structure of a dual-layer optical encryption fluorescent polymer waveguide chip is proposed as illustrated in Fig. 2b. The cross-sectional and top-view structure of the dual-layer waveguide device are given in Figs. 2c and d, respectively. The TCBzC/SU-8 is selected as the bottom layer for optical information transmission and the abbreviation of Jilin University (JLU), with a 532 nm wavelength optical source used as the signal light. A self-synthesized P(MMA-co-GMA) polymer with high transparency and thermal stability is chosen for the upper, lower, and buffer layers of the dual-layer waveguide device. And then the TCNzC/SU-8 was used as the top waveguide layer material for the optical information transmission and the logo of Jilin University. By emitting pumping light at different power intensities at the top of the waveguide device, different types of fluorescent material doped optical waveguide will absorb the pumping light and produce gains.

Based on the proposed structure of the dual-layer optical encryption waveguide, we have optimized the detailed structural sizes of the waveguide as given in Fig. 3. The cross-sectional size of the polymer waveguide core layer is $5 \times 7\,\mu m^2$ as displayed in Figs. 3a and b,

so that the overlapping integral factor of the fundamental modes for the pumping light and signal source could reach more than 99%. The distance between the buffer layer and two waveguide layers is 4 μm, which allows the optical field to be better confined in the core layer. To couple the optical waveguide device with a commercial fiber array (Shijia Photons, G657A-1m-FC/APC), the horizontal spacings between the input and output waveguide channels for top and bottom layers are defined as 127 μm. The length of the bending waveguide is 1500 μm, and the length of the straight waveguide in the tangential section is 1150 μm. The waveguides of top and bottom layers are tangent in the vertical direction to ensure that the waveguides in both layers could absorb the pumping light energy sufficiently. After finishing the 3D structural design optimization of the dual-layer encryption waveguide device, we fabricated the device by mature semiconductor process techniques.

## Fabrication and testing of dual-layer optical encryption waveguide chip

The designed dual-layer optical encryption fluorescent polymer waveguide chip could be fabricated by UV direct written technique as given in Fig. 4. The fabrication process of the bottom waveguide layer chip was shown in Fig. 4a–f, and for the top waveguide layer was shown in Fig. 4g–j. A standard silicon wafer with the SiO$_2$ buffer layer of 5 μm thickness was immersed in acetone and cleaned with an ultrasonic cleaner for 5 min. Then the organic solvent was removed by means of

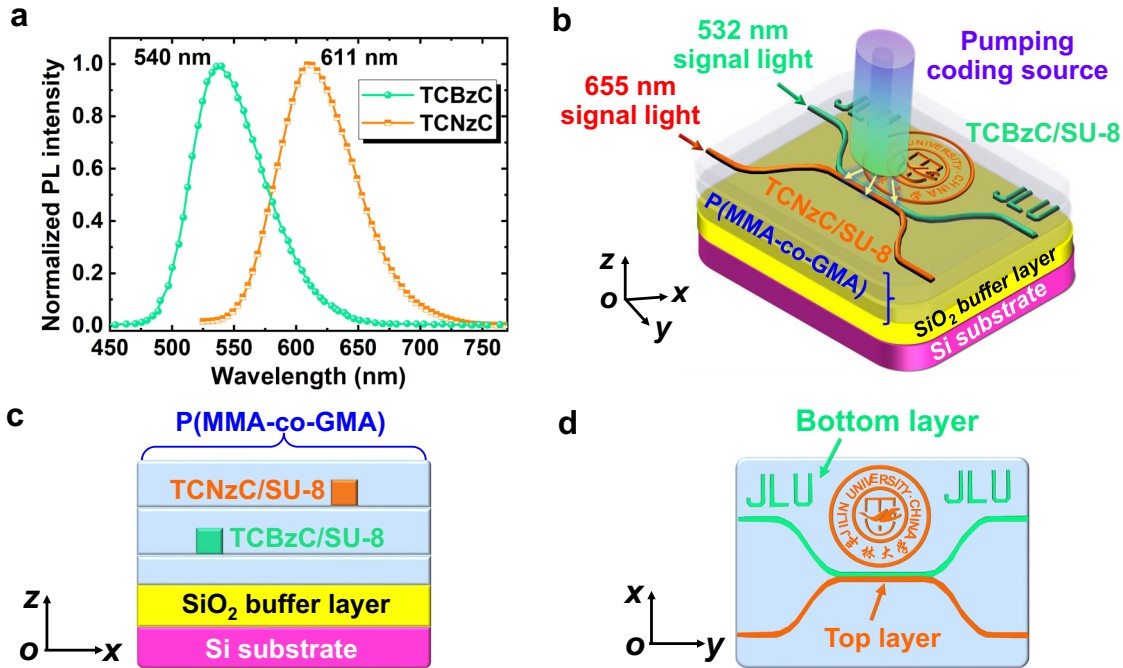

**Fig. 2 | Structural design of a dual-layer encryption optical waveguide device. a** The normalized PL intensity spectra of TCBzC and TCNzC fluorescent material; **b** 3D schematic diagram of the dual-layer waveguide; **c** cross-sectional structure of the waveguide output port; **d** top view structure of the waveguide device.

ultrasound in iso-propyl alcohol (IPA) for 5 min. After cleaning with deionized water and blow-drying with nitrogen gun, the Si substrate was dried in an oven and heated at 120 °C for 30 min as illustrated in Fig. 4a. After drying, the SiO₂ substrate was prepared for the next process. Firstly, the 10 μm thickness P(MMA-co-GMA) lower cladding was spinning-coated on the SiO₂ substrate as given in Fig. 4b, and then the TCBzC/SU-8 was rotated on the lower cladding layer as shown in Fig. 4c. The thickness of the waveguide core layer was formed into 5 μm by controlling the rotating speed as 3000 r/min for 30 s. The fluorescent polymer thin film was prebaked on a heating plate at 60 °C for 10 min, and then up to 90 °C for 20 min. After cooling to room temperature at 25 °C, the polymer thin film was exposed for 7 s with the lithography machine (ABM/6/350) in Fig. 4d. Then with post-baking (65 °C for 10 min, and 95 °C for 20 min), the epoxy groups in SU-8 were cross-linked at the exposed position, which can improve thermal stability. Finally, the UV written waveguide chip was immersed in photoresist developer (propylene glycol methyl ether acetate (PGMEA)) for 15 s as given in Fig. 4e. After that, the polymer thin film region without epoxy cross-linked was dissolved. It was then placed on the heating plate for hard-baking at a temperature of 120 °C for 30 min. Next, the bottom waveguide layer covered with P(MMA-co-GMA) as a buffer layer (at a speed of 3000 rpm @ 20 s) was shown in Fig. 4f. The single-layer optical encryption waveguide chip was realized as illustrated in Fig. 4a–f. For the top layer polymer waveguide chip, the fabrication process was similar to above. The TCNzC/SU-8 polymer thin film was spin-coated on the P(MMA-co-GMA) buffer layer at the speed of 3000 rpm@20 s as given in Fig. 4g. The positions of the dual-layer waveguide sidewalls were aligned during the exposure fabrication process. And the detailed operating process was the same as Fig. 4d–f. Finally, the dual-layer optical encryption fluorescent polymer waveguide chip was achieved as given in Fig. 4a–j.

The dual-layer optical waveguide device was fabricated with the above process. The top view of the waveguide structure was captured as given in Fig. 5a. The structures of the dual-layer waveguide, the university logo, and JLU abbreviation doped with the different fluorescent polymers were clear and intact. The cross-sectional structure of the dual-layer waveguide was collected by a scanning electron

microscopy (SEM, JSM-7900F) as shown in Fig. 5b, c. It can be seen that a clear demarcation line exists between the waveguide core and cladding layer materials, demonstrating that there is no miscibility phenomenon in different waveguide layers. In addition, the wet-etching method allows for a better shaped waveguide core layer structure. The waveguide core layer size was 7.3 × 5 μm², which is consistent with the design dimensions. It shows that shaped core waveguide layer structures can be realized well by spin-coating and wet-etching process. This method eliminates the need for epitaxial growth techniques, dry-etching, and mask preparation, which not only greatly simplifies the fabrication process of multilayer waveguide, but also helps to reduce manufacturing costs and accelerate the speed of waveguide device fabrication. We used a 365 nm point light source to irradiate the fabricated dual-layer waveguide chip to verify the fluorescence effect, as shown in Fig. 5d. A partially enlarged photograph of the device was given in Fig. 5e. It can be clearly observed that the different fluorescent polymers can emit green and orange light, respectively, while the bottom and top layers of the aligned cross markers showed a yellow color due to the overlay of the two colors. After finishing the fabrication of the dual-layer waveguide device, the optical properties and the dynamic modulation response were then tested in order to verify its optical pulse-code modulation communication encryption performances.

An optical pulse-code modulation coupling testing system was built as illustrated in Fig. S1. For the analysis of the optical gain characteristics of the dual-layer waveguide chip based on continuous-wave (CW) pumping light, we can measure the output spectra at different power pumping light with the fiber optic spectrometer (FX2000) connected to a server computer, and then the gains at different pumping light intensities could be calculated. When analyzing the optical encryption communication characteristics of the waveguide chip, the 405-nm laser was modulated as a pulse-code pumping light source. The output optical response signal from the dual-layer encryption optical waveguide device was then coupled into the photoelectric detector and converted into an electrical signal, which can be displayed on a digital oscilloscope (DS4024) for encryption modulation wave and response time testing. At the same time, the original

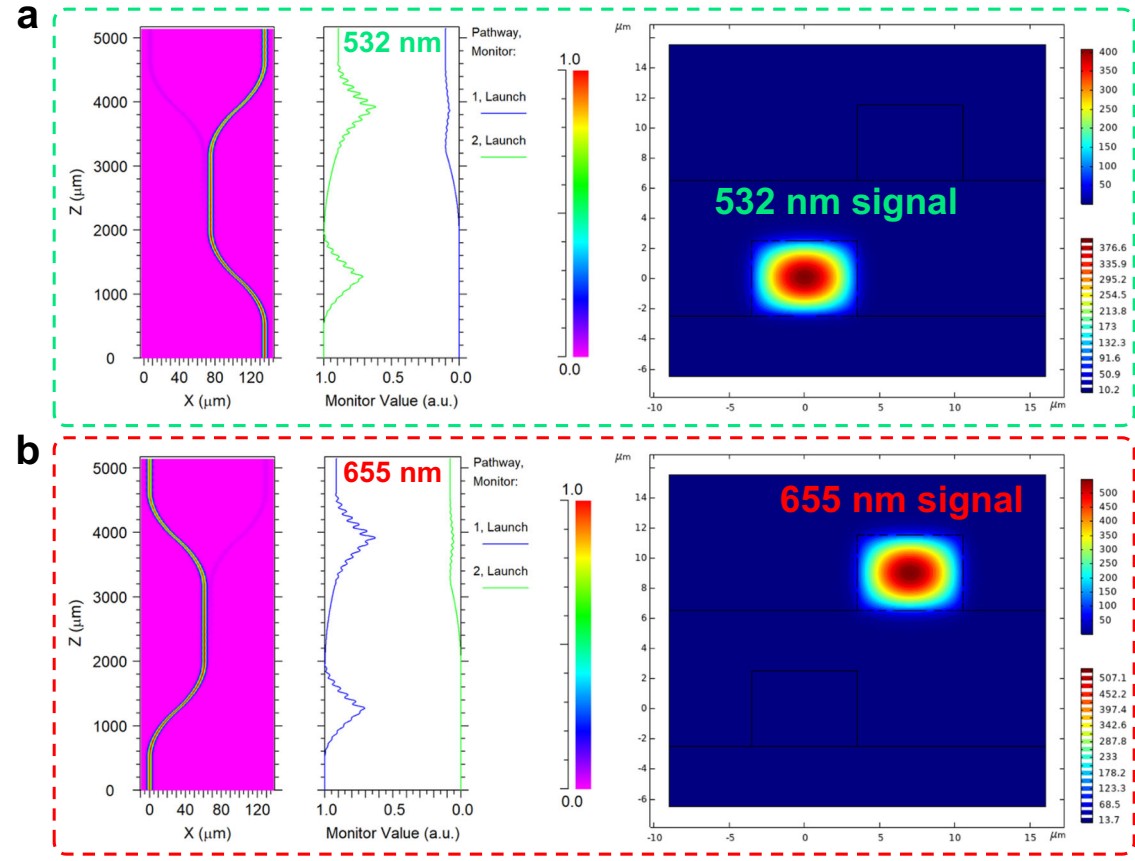

**Fig. 3 | Structural design of the dual-layer optical encryption fluorescent polymer waveguide chip. a** The optical field distributions of the bottom waveguide layer in the horizontal and cross-sectional directions at 532 nm wavelength signal light; **b** the optical field distributions of the top waveguide layer in the horizontal and cross-sectional directions at 655 nm wavelength signal light.

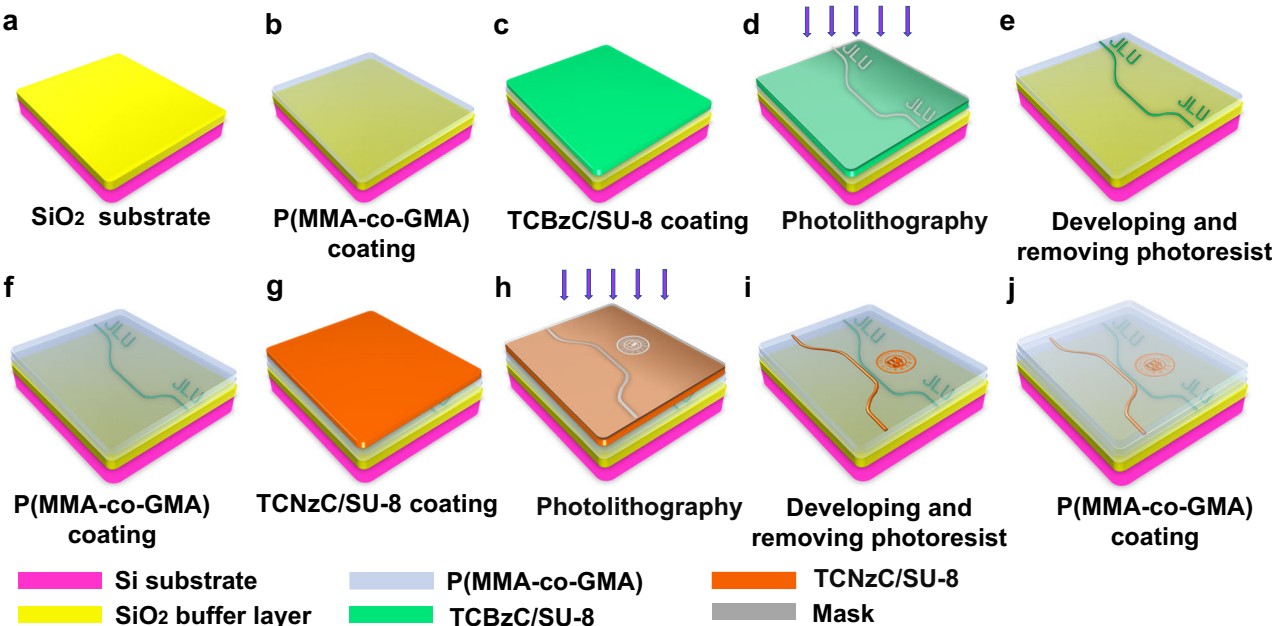

**Fig. 4 | The fabrication process of the dual-layer encryption fluorescent optical waveguide chip. a** Cleaning the silicon substrate; **b** spin-coating the lower cladding layer; **c** spin-coating of TCBzC/SU-8 core layer; **d** exposure of the bottom waveguide layer device; **e** wet-etching for the waveguide; **f** spin-coating the buffer layer; **g** spin-coating of TCNzC/SU-8 top layer; **h** exposure of the top waveguide layer device; **i** wet-etching for the waveguide; **j** spin-coating the upper cladding layer.

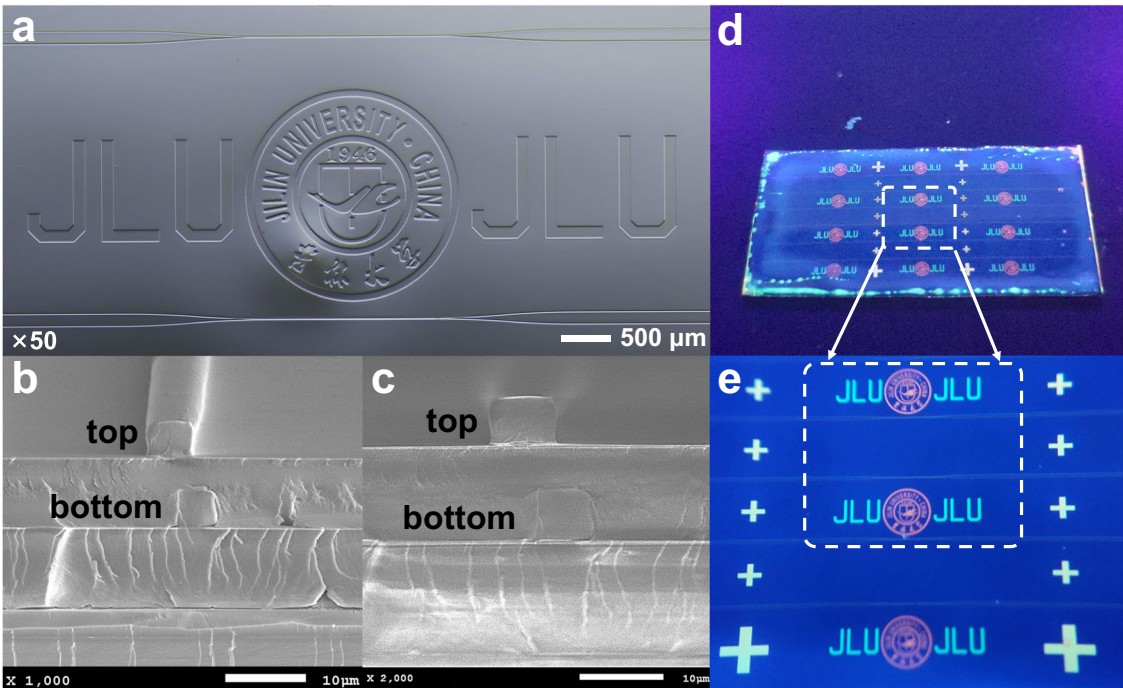

**Fig. 5 | Structural characterization and fluorescence emission of the fabricated dual-layer encryption waveguide. a** The microscope captured image of the whole waveguide structure; (**b**) and (**c**) are SEM images of the waveguide cut-surface; (**d**) and (**e**) are fluorescence emission images of the dual-layer waveguide chip under a 365 nm point source.

frequency pulse-code modulation square-wave generated by the digital signal generator will be selected as a reference signal.

As captured in Fig. 6a, b, the photographs were taken with vertical and horizontal CCD microscope cameras by coupling the 532 and 655 nm signal lights into the dual-layer waveguide chip with 50 mW power intensity of 405 nm pumping light as the external irradiation source. It can be noticed from Fig. 6a that the both signal lights were able to realize a stable optical beam transmission at different waveguide layers. At the same time, the university logo and the JLU abbreviation on both waveguide layers could emit the corresponding orange and green light, respectively. Furthermore, we can observe that with the same input optical power intensity, the output optical power intensity of the 532 nm signal light in the bottom TCBzC/SU-8 layer (right) is higher than that of the 655 nm signal light in the top TCNzC/SU-8 layer (left), when the pumping light was applied to the waveguide chip in Fig. 6b. The main reason for the detectable intermittent bright spots in Fig. 6b should be due to the light leakage phenomenon. It might be caused by the actual refractive index perturbation from the thermal stress change between the upper and lower cladding in fabrication process. Further, we measured the relative gains produced by different external pumping light power intensities for two different wavelengths of signal light. The relative gain is defined as:

$$Gain(\text{dB}) = 10 \lg\left(\frac{P_{out}^{p}}{P_{out}}\right) \qquad (1)$$

where $P_{out}^{p}$ is the power of the waveguide output signal light when 405 nm pumping light applied, and $P_{out}$ is the optical power intensity of the output signal without pumping light applied. When the input signal light wavelength was 532 nm, we have measured the relative gain of the bottom TCBzC/SU-8 waveguide layer with the pumping light power applied, when the optical power of the signal light is 30 mW, as summarized in Table 1. And the illustration drawn from the data was

shown in Fig. 6c. It can be observed that when the power of the input signal light is constant, the relative gain of the measured waveguide gradually increased with the increasing in the power intensity of the external pumping light, which showed a linear growth. When the pumping light power was 100 mW, the maximum relative gain at 532 nm wavelength signal source was 5.71 dB. When the signal light wavelength was 655 nm, the increasement amplitude in relative gain decreased and flatten out as the applied external pumping light power grown. The maximum relative gain at 655 nm wavelength was 5.34 dB when the pumping light power was 100 mW, as summarized in Table 1. And the illustration from the data was shown in Fig. 6d. It can be demonstrated that the relative gain of the TCBzC/SU-8 waveguide was slightly higher than that of the TCNzC/SU-8 waveguide when the optical power of 405 nm wavelength external pumping light was equal.

The optical pulse-code modulation performance can be carried out by utilizing the feature that both of fluorescent polymer top and bottom waveguides can generate relative gain at 405 nm wavelength external pumping light. The photonic plaintext digital information can be simultaneously generated by both bottom TCBzC/SU-8 waveguide layer for external optical fiber network and top TCNzC/SU-8 waveguide layer for internal optical fiber standby network. When message codes as [1000] and [1100] are output at a frequency of 250 Hz, respectively (yellow square-wave), the corresponding blue square-waves on the digital oscilloscope from the bottom TCBzC/SU-8 waveguide layer are obtained as shown in Fig. 6e, f, respectively. And the rise and fall time were measured to be 260 and 300 µs, respectively. As given in Fig. 6g, h, message codes as [0110] and [0111] are generated at the frequency of 250 Hz, respectively (yellow square-wave). The corresponding blue square-waves on the digital oscilloscope are received from the top TCNzC/SU-8 waveguide layer. And the rise and fall time were measured to be 270 and 300 µs, respectively. The rapid responsive time and comprehensive message transmission capacity demonstrated the feasibility of the waveguide device for optical pulse-code modulation encryption communication.

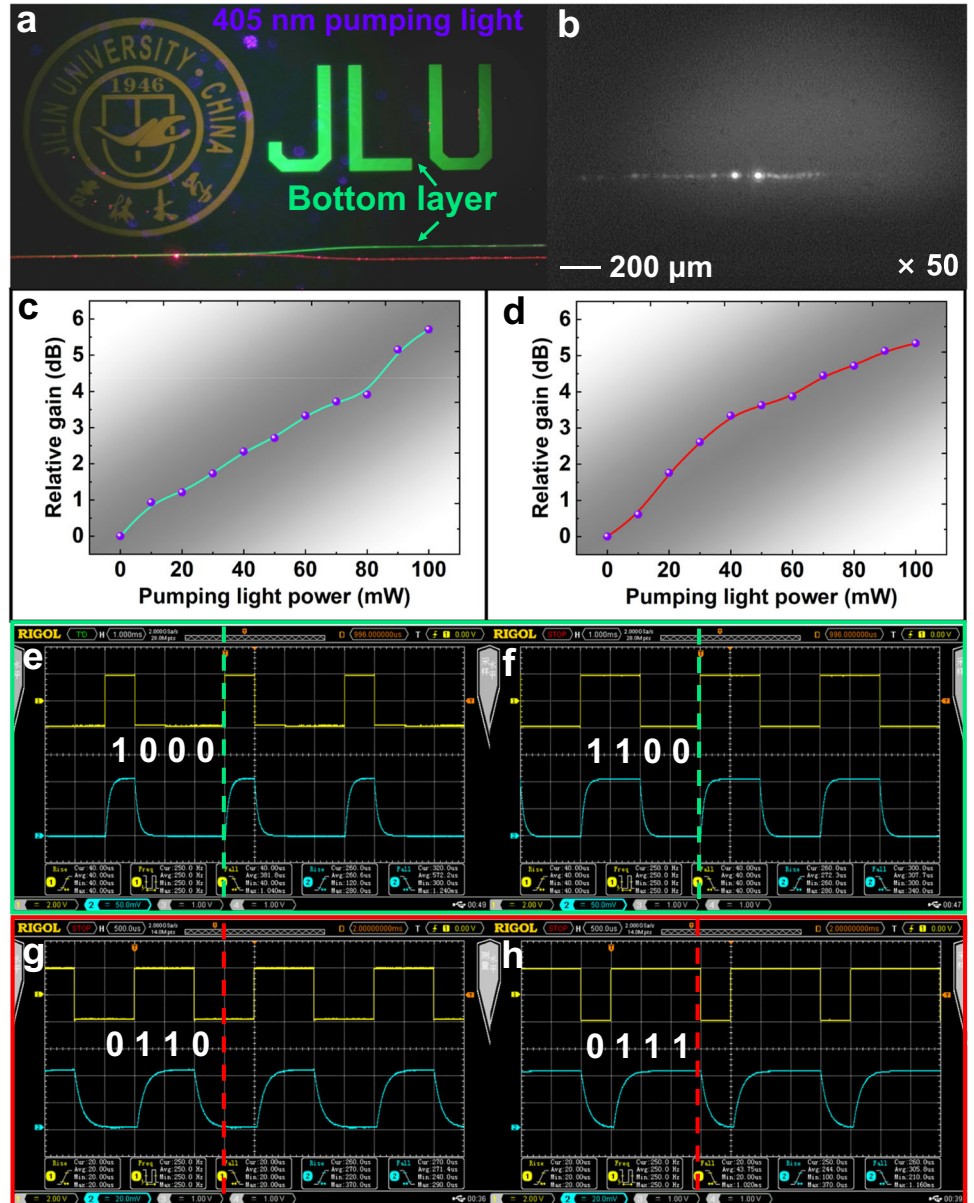

**Fig. 6 | The analysis of the optical response properties of the dual-layer waveguide device with 405 nm wavelength pumping light excitation. a** The top-view microscope when coupling 532 and 655 nm signal lights with 405 nm wavelength light pumping in; **b** the output optical fields of both signal sources (×50); (**c**) and (**d**) are the curves of relative gain versus pumping light power intensity variation for 532 and 655 nm wavelength signal source; (**e**) and (**f**) are the input/output response square-wave with pulse-code modulation for 532 nm wavelength signal source at a frequency of 250 Hz; (**g**) and (**h**) are the input/output response square-wave with pulse-code modulation for 655 nm wavelength signal source at a frequency of 250 Hz.

**Table 1 | The relative gains of the bottom TCBzC/SU-8 waveguide and top TCNzC/SU-8 waveguide excited by 405 nm wavelength pumping light**

| WL | $\lambda_S$ (nm) | $\lambda_P$ (nm) | PLP (mW) | | | | | | | | | |
|---|---|---|---|---|---|---|---|---|---|---|---|---|
| | | | 10 | 20 | 30 | 40 | 50 | 60 | 70 | 80 | 90 | 100 |
| TCBzC/SU-8 | 532 | 405 | RG (dB) | | | | | | | | | |
| | | | 0.93 | 1.21 | 1.73 | 2.34 | 2.71 | 3.33 | 3.72 | 3.91 | 5.15 | 5.71 |
| TCNzC/SU-8 | 655 | 405 | 0.61 | 1.76 | 2.61 | 3.34 | 3.63 | 3.87 | 4.45 | 4.72 | 5.13 | 5.34 |

*WL* waveguide layer, $\lambda_S$ signal light wavelength, $\lambda_P$ pumping light wavelength, *PLP* pumping light power, *RG* relative gain.

## Discussion

In this section we discuss how our dual-layer encryption waveguide device reacts to protect the transmission of encryption message when it is eavesdropped. It is crucial that when any optical encryption chip is inserted into optical fiber communication system, it is not desired to bring any extra optical loss during long-distance information transferring network so that to avoid the potential eavesdropper detection. In experiment, initial 30 mW of external 405 nm wavelength pumping light is constantly provided to the bottom TCBzC/SU-8 waveguide layer to compensate the insertion loss of the optical chip for 532 nm

wavelength signal light. When the optical encryption channel is utilized to ensure the security of information transmission, initial 30 mW of external 532 nm wavelength pumping light is continuously supplied to the top TCNzC/SU-8 waveguide layer to balance the insertion loss for the optical chip for 655 nm wavelength signal light. 30 mW is set as the constant operating pumping optical power for the optical encryption chip. The proposed loss-compensation technique for optical encryption chips will be beneficial to guarantee the security of internet communication for preventing the detection from the potential eavesdropper. In actual experiment, the approach to monitor the optical transmission loss change in the optical fiber system is adopted to detect the eavesdropping[40]. The power of the optical signal is a typical and significant parameter, so it could reflect the signal loss by eavesdropping. In Fig. 7a, the scheme to detect eavesdropping for our proposed optical encryption waveguide chip is described. The optical fiber coupler with 95/5 power splitting ratio is used to demonstrate detect eavesdropping case. 5% signal optical power as eavesdropping influence is measured by the eavesdropper port and

95% signal optical power normally transmitted is tested by the receiver port. To detect the eavesdropping of an intruder, as given in Fig. 7b, the bottom TCBzC/SU-8 waveguide inputted with 532 nm signal wavelength is pumped externally by different light powers as 30, 50, 70, and 100 mW at 405 nm wavelength, respectively. The input signal power is set at 0 dBm (1 mW) and the output signal powers from both the eavesdropper and receiver ports are measured at intervals of 20 s for five times by optical power meter (Thorlabs, PM100D). Specially, the output signal powers from the receiver port is retested and collected three times. Depending on the contrast data of the output signal powers between both eavesdropper and receiver ports, it could be found that average 0.13 dB power difference close to 5.84% power splitting ratio is obtained with 30, 50, 70, and 100 mW pumping light power. The detecting accuracy could reach 95.2%. It could be alarmed that there might be the eavesdropper by this method. When the intruder is discovered, some pseudo-code information will be used to confuse eavesdropper by the bottom TCBzC/SU-8 waveguide layer while the top TCNzC/SU-8 waveguide layer is utilized to transfer true

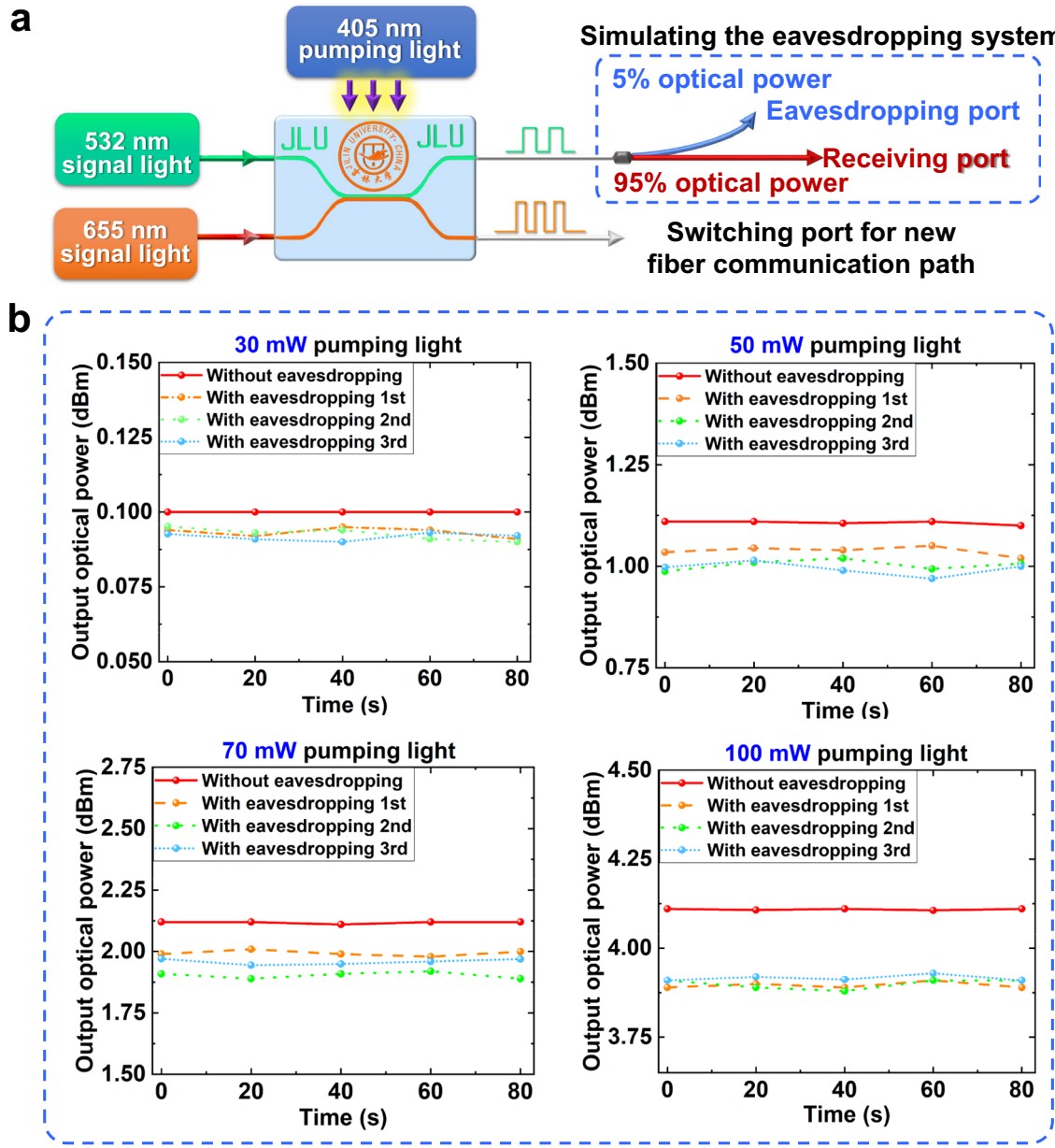

**Fig. 7 | The message transmission protecting system and measuring method of the dual-layer encryption waveguide device. a** The scheme to detect eavesdropping for the waveguide chip; **b** the output signal powers from the receiver port pumped externally by 30, 50, 70, and 100 mW at 405 nm wavelength, respectively.

information data. 655 nm wavelength signal light pumped by 532 nm wavelength modulating light in the top TCNzC/SU-8 waveguide layer will be transferred into another fiber communication path. The variation of optical operating frequency for both signal and pumping light could also be helpful to ensure the network information security and avoid eavesdropping.

If an external eavesdropper tries to steal the optical information transmitted, we would still use the external optical fiber network to transmit pseudo-coding information for confusing the eavesdropper in the bottom TCBzC/SU-8 waveguide layer. Meanwhile, we would switch the 405 nm wavelength pumping light to 532 nm wavelength in optical fiber standby network to transfer the alarming information to the receiver by the top TCNzC/SU-8 waveguide layer. Through modifying both the external pumping wavelength and driving frequency of the pulse-code modulation, transmitters can ensure the transmission of vital information securely. As shown in Fig. 8a, when the pumping light was 532 nm length, the transmission of the image and optical signal through the bottom TCBzC/SU-8 waveguide layer was hidden and only the 655 nm signal light was transmitted in the top TCNzC/SU-8 waveguide layer. This method could protect the image information and encryption data in the TCBzC/SU-8 waveguide layer. We used a CCD camera to capture the output light field of the 655 nm signal light as given in Fig. 8b, and can see that as the external pumping light power increased, the power energy of the output light field was also enhanced. When the pumping light power was 100 mW, the maximum relative gain at 655 nm wavelength was 7.45 dB. The relative gain of the

TCNzC/SU-8 waveguide was summarized in Table 2. As depicted in Fig. 8c–e, we have carried out the pseudo-code information modulation by applying 405 nm pumping light at a frequency of 200 Hz through the bottom TCBzC/SU-8 waveguide layer to confuse eavesdropper. The pseudo-code states as [1000], [1001], and [1010] were selected for the testing, respectively. As given in Fig. 8f–h, the true pulse-code information modulation by applying 532 nm pumping light at a frequency of 500 Hz was generated in optical fiber standby network. The pulse-code states as [1100], [0001], and [1110] were measured for the testing by the top TCNzC/SU-8 waveguide layer, respectively. The response time was measured to be 264 μs. The experimental results demonstrated that the technique has favorable information encryption performance and can effectively ensure that the authorized receiving users get encryption information in simultaneous real time. It allows the dual-layer optical waveguide chip to be used for graphic encryption and the control of data signal transmission in optical communication systems. This proposed design has valuable applications in all-optical secure communications.

## Outlook

In summary, a dual-layer optical encryption fluorescent polymer waveguide chip based on optical pulse-code modulation technique is proposed for optical encryption communication. TCBzC and TCNzC were doped into epoxy cross-linking SU-8 polymer as a gain medium. Through modifying both the external pumping wavelength and driving frequency of the pulse-code modulation, transmitters can ensure

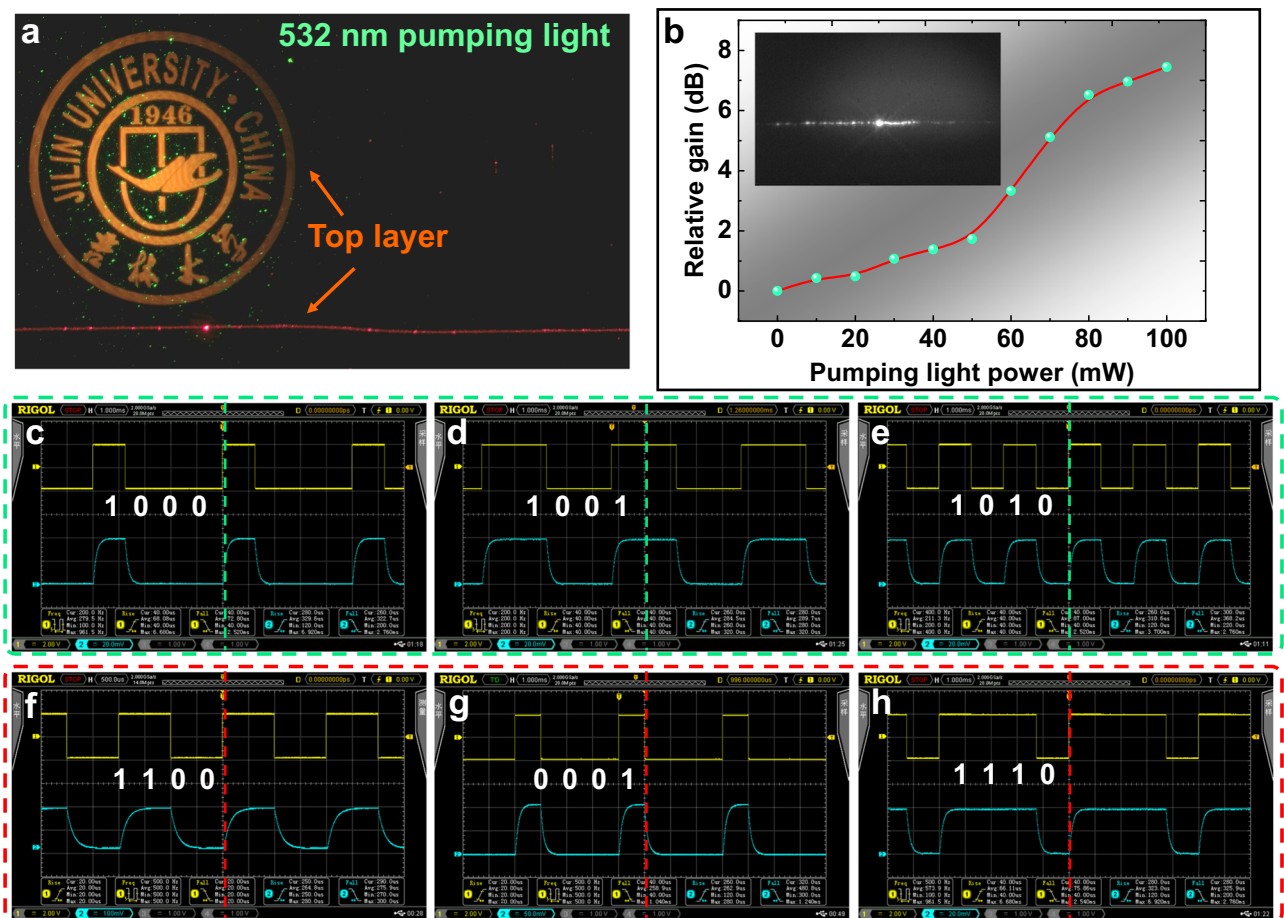

**Fig. 8 | The analysis of the optical response properties while being eavesdropped. a** The top-view microscope when coupling 655 nm signal light with external 532 nm wavelength light pumping in; **b** the output optical field of the signal source and the curve of relative gain versus different pumping light power intensity; (**c**) to (**e**) are the input/output response square-wave with pulse-code modulation for 532 nm wavelength pseudo-code information; (**f**) to (**h**) are the input/output response square-wave with pulse-code modulation for 655 nm wavelength pulse-code information.

**Table 2 | The relative gains of the top TCNzC/SU-8 waveguide excited by 532 nm wavelength pumping light**

| WL | $\lambda_S$ (nm) | $\lambda_P$ (nm) | PLP (mW) | | | | | | | | | |
|---|---|---|---|---|---|---|---|---|---|---|---|---|
| | | | 10 | 20 | 30 | 40 | 50 | 60 | 70 | 80 | 90 | 100 |
| TCNzC/SU-8 | 655 | 532 | RG (dB) | | | | | | | | | |
| | | | 0.43 | 0.49 | 1.07 | 1.38 | 1.73 | 3.33 | 5.12 | 6.52 | 6.96 | 7.45 |

*WL* waveguide layer, *$\lambda_S$* Signal light wavelength, *$\lambda_P$* Pumping light wavelength, *PLP* pumping light power, *RG* relative gain.

the transmission of vital information securely. When the plaintext information is transmitted securely in bottom waveguide layer, the relative gain of 532 nm wavelength signal source at 100 mW of 405 nm pumping light was 5.71 dB, and the response time was measured to be 260 μs. If the plaintext transmission is eavesdropped, the wavelength of the external pumping light is switched, and the receiver will get warning commands of ciphertext information in the top waveguide layer. This technique provides significant promising developments for realizing Internet optical encryption communications in education, commercial transactions, and national military security.

## Methods

### Waveguide materials preparation
The fluorescent small molecule oligomers green-light TCBzC and red-light TCNzC with different luminescent rigid cores (2,1,3-benzothiadiazole for TCBzC and 2,1,3-naphthalenediazole for TCNzC) in the molecules were selected as fluorescent light-emitting materials, so that green and red-light emission could be achieved, respectively. In the experiment, TCBzC and TCNzC with the mass fraction of 5 wt‰ were doped in the epoxy cross-linking SU-8 polymer, respectively. For the cladding layer material P(MMA-co-GMA), we synthesized it with methyl methacrylate (MMA) and propylene oxide methacrylate (GMA) by copolymerization.

### Fabrication and characterization
The refractive index of the waveguide fabrication material is measured by the ellipsometer (SPEL M-2000VI, America). The waveguide core layer structure was fabricated by the lithography machine (ABM/6/350). The cross-sectional structure of the dual-layer waveguide was collected by the scanning electron microscopy (SEM, JSM-7900F).

### Measurements
The 532-nm and 655-nm wavelength laser sources (MDL-III series) were chosen as the signal source coupled to the waveguide chip, respectively. A 405-nm wavelength laser was directly used as an external pulse-code modulation pumping light, which was generated pulse-code optical signal by the digital signal generator (SP1642B) loaded on.

## Data availability
The data that support this study is available from the corresponding author upon request.

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

## Acknowledgements

The authors would like to thank Dr. Tonghe Sun. She provided valuable help with the experiment testing. This work was supported by the National Key Research and Development (R&D) Program of China (2019YFB2203001) and the National Natural Science Foundation of China (NSFC, No. 62171195).

## Author contributions

C.W. and C.C. designed this work and completed the experiments; T.F. and T.Z. synthesized the materials and drew the figures; D.Z. assisted in lighting experiments with X.S., A.C., H.L., and X.Z.; C.W., T.F., and J.Y. wrote the paper; all authors participated in the analysis of experimental data and discussion of the results.

## Competing interests

The authors declare no competing interests.
