## [Peer Review File · Nature Communications]

Dual-layer optical encryption fluorescent polymer waveguide chip based on optical pulse-code modulation techniqueReviewer #1 (Remarks to the Author):

In the manuscript "Dual-layer optical encryption fluorescent polymer waveguide chip based on optical pulse-code modulation technique," Wang et al. present a dual-layer optical encryption fluorescent polymer waveguide chip that uses an optical pulse-code modulation technique for optical encryption communication. The authors propose the interesting idea of using an external light source to modulate two signal light sources (532 and 655 nm) to display encrypted information in the vertical direction and transmit digital information in the horizontal direction through the fluorescent waveguide device with dual gain layers. By switching the wavelength and frequency of the external pumping light, transmitters can initiate or interrupt the transmission of sensitive information.

The manuscript is well-written and technically sound with clear illustrations. However, a major concern is the insufficient discussion of how the system would detect eavesdropping. The process is not explained in detail and is unclear.

There are also a few questions that need to be addressed before publication:

1. The refractive index of P(MMA-co-GMA) is mentioned in line 107, but the refractive indices of the solutions are only shown in line 144, despite being mentioned earlier.
2. The authors mention external light source wavelengths of 365 nm and 405 nm, and it can be confusing. A clearer explanation of the differences between them and the reason for choosing 405 nm over 365 nm, despite showing the response to 365 nm in Fig. 2a, would be helpful.
3. Line 149 mentions commercial fiber arrays without references, which should be included.
4. A discussion of potential losses during long-distance information transfer, given the proposed use in internet communications, would be beneficial.

In conclusion, the manuscript presents an intriguing idea for secure communication, but some of the comments need to be addressed before it can be considered for publication. The most crucial issue is the lack of explanation regarding how the system would detect an intruder and communicate it to the external light source.

In this manuscript, the authors report a dual-layer optical encryption fluorescent polymer waveguide chip based on optical pulse-code modulation technique. The theoretical background and characterizations are relatively comprehensive. This work is interesting, and the manuscript may be recommended for publication after following issues have been addressed successfully.

1. For Figure 2a, the vertical coordinate seemed to be normalized for PL intensity. The author should carefully check this point. In addition, the author should provide the absorbance spectra of TCBzC and TCNzC. I am wondering that whether is there a fluorescence energy resonance transfer (FERT) occurred between TCBzC and TCNzC, which also might their resulting waveguide performance.
2. In this work, why the author choose 655nm light sources as signal sources while the fluorescence emission of TCNzC is at 610 nm?
3. As in Figure 7b, what is the main reason for the detectable intermittent bright spots? whether it is caused by light leakage phenomenon, or due to the fluorescence molecules dispersed within polymer matrix? Meanwhile, in Figure 7b, the author should add the bar scale.
4. The captions of the Figure 8c and 8d are absent.
5. In this paper, the corresponding characterization of the optical-loss coefficient of optical waveguide materials should be characterized.

Reviewer's comments (blue) and our answers (black)

Dear Reviewer 1:

We would like to express our gratitude to you for helping us to improve the manuscript. Gratefully thank you for the valuable comments. The main revisions as well as the explanations are shown in the following:

In the manuscript "Dual-layer optical encryption fluorescent polymer waveguide chip based on optical pulse-code modulation technique," Wang et al. present a dual-layer optical encryption fluorescent polymer waveguide chip that uses an optical pulse-code modulation technique for optical encryption communication. The authors propose the interesting idea of using an external light source to modulate two signal light sources (532 and 655 nm) to display encrypted information in the vertical direction and transmit digital information in the horizontal direction through the fluorescent waveguide device with dual gain layers. By switching the wavelength and frequency of the external pumping light, transmitters can initiate or interrupt the transmission of sensitive information.

The manuscript is well-written and technically sound with clear illustrations. However, a major concern is the insufficient discussion of how the system would detect eavesdropping. The process is not explained in detail and is unclear.

There are also a few questions that need to be addressed before publication:

1. The refractive index of P(MMA-co-GMA) is mentioned in line 107, but the refractive indices of the solutions are only shown in line 144, despite being mentioned earlier.

Thanks for your comment. According to your suggestion, we moved the part of the refractive indices of the waveguide core layer materials to the "Materials and methods-Fluorescent gain polymer waveguide materials" section as follows:

"The refractive index of the self-synthesized P(MMA-co-GMA) film is 1.490 measured by an ellipsometer (SPEL M-2000VI, America) from 500 to 700 nm wavelength, which has a minor birefringence and can be adjusted in a wide range. The varied curves for refractive index (n) and extinction coefficient (k) in Vis-NIR wavelength region for TCBzC/SU-8 and TCNzC/SU-8 waveguide materials are measured by the ellipsometer. The refractive index of bottom layer waveguide material TCBzC/SU-8 is 1.597 at 532 nm wavelength and for top layer waveguide material TCNzC/SU-8 is 1.587 at 655 nm wavelength as given in **Figure 1c**."

2. The authors mention external light source wavelengths of 365 nm and 405 nm, and it can be confusing. A clearer explanation of the differences between them and the reason for choosing 405 nm over 365 nm, despite showing the response to 365 nm in Fig. 2a, would be helpful.

Thanks for your suggestion. Instead of 365 nm wavelength, 405 nm wavelength light is used as the external optical pumping source to modulate the actual photonic chip. The main reason is that 365 nm wavelength as ultraviolet (UV) light might cause photo-bleaching phenomenon for the dye oligomers (TCBzC and TCNzC) in polymer waveguide. This would influence the operating performance and stability of the proposed waveguide chip. Contrast to 365 nm

wavelength UV light, 405 nm wavelength as visible light has enough photon energy while hardly result in damage to the dye oligomers in polymer waveguide. Therefore, 365 nm UV wavelength light is chosen to analyze PL characteristic of the dye oligomers (TCBzC and TCNzC) and 405 nm visible wavelength light is used to pump the actual waveguide chip by optical pulse-code modulation technique. The detailed explanation is added into the revised manuscript as follows:

“Compared to 365 nm ultraviolet (UV) wavelength light chosen to analyze PL characteristic of the dye oligomers (TCBzC and TCNzC), 405 nm visible wavelength light is used to pump the actual waveguide chip. The main reason is that 365 nm UV wavelength light might cause photo-bleaching phenomenon for the dye oligomers in polymer waveguide. Contrast to 365 nm wavelength UV light, 405 nm visible wavelength has enough photon energy while hardly result in damage to the dye oligomers in polymer waveguide. Therefore, 405 nm visible wavelength light is used to pump the waveguide chip by optical pulse-code modulation technique in actual experiment.”

3. Line 149 mentions commercial fiber arrays without references, which should be included.

Thanks for your suggestion. The commercial fiber array (Shijia Photons, G657A-1m-FC/APC) is used to couple with the optical waveguide device. It has been added to the revised manuscript as follows:

“To couple the optical waveguide device to a commercial fiber array (Shijia Photons, G657A-1m-FC/APC), the horizontal spacings between the input and output waveguide channels for top and bottom layers are defined as 127 μm .”

4. A discussion of potential losses during long-distance information transfer, given the proposed use in internet communications, would be beneficial.

Thanks for your comment. It is crucial that when any optical encryption chip is inserted into optical fiber communication system, it is not desired to bring any extra optical loss during long-distance information transferring network so that to avoid the potential eavesdropper detection. In experiment, initial 30 mW of external 405 nm wavelength pumping light is constantly provided to the bottom TCBzC/SU-8 waveguide layer to compensate the insertion loss of the optical chip for 532 nm wavelength signal light. When the optical encryption channel is utilized to ensure the security of information transmission, initial 30 mW of external 532 nm wavelength pumping light is continuously supplied to the top TCNzC/SU-8 waveguide layer to balance the insertion loss for the optical chip for 655 nm wavelength signal light. 30 mW is set as the constant operating pumping optical power for the optical encryption chip. The proposed loss-compensation technique for optical encryption chips will be beneficial to guarantee the security of internet communication for preventing the detection from the potential eavesdropper. The detailed explanation is added into the revised manuscript as follows:

“It is crucial that when any optical encryption chip is inserted into optical fiber communication system, it is not desired to bring any extra optical loss during long-distance information transferring network so that to avoid the potential eavesdropper detection. In experiment,

initial 30 mW of external 405 nm wavelength pumping light is constantly provided to the bottom TCBzC/SU-8 waveguide layer to compensate the insertion loss of the optical chip for 532 nm wavelength signal light. When the optical encryption channel is utilized to ensure the security of information transmission, initial 30 mW of external 532 nm wavelength pumping light is continuously supplied to the top TCNzC/SU-8 waveguide layer to balance the insertion loss for the optical chip for 655 nm wavelength signal light. 30 mW is set as the constant operating pumping optical power for the optical encryption chip. The proposed loss-compensation technique for optical encryption chips will be beneficial to guarantee the security of internet communication for preventing the detection from the potential eavesdropper.”

5. In conclusion, the manuscript presents an intriguing idea for secure communication, but some of the comments need to be addressed before it can be considered for publication. The most crucial issue is the lack of explanation regarding how the system would detect an intruder and communicate it to the external light source.

Thanks for your comment. In our experiment, the approach to monitor the optical transmission loss change in the optical fiber system is adopted to detect the eavesdropping.^[40] The power of the optical signal is a typical and significant parameter, so it could reflect the signal loss by eavesdropping. In **Figure 8a**, the scheme to detect eavesdropping for our proposed optical encryption waveguide chip is described. The optical fiber coupler with 95/5 power splitting ratio is used to demonstrate detect eavesdropping case. 5% signal optical power as eavesdropping influence is measured by the eavesdropper port and 95% signal optical power normally transmitted is tested by the receiver port. To detect the eavesdropping of an intruder, as given in **Figure 8b**, the bottom TCBzC/SU-8 waveguide inputted with 532 nm signal wavelength is pumped externally by different light powers as 30, 50, 70, and 100 mW at 405 nm wavelength, respectively. The input signal power is set at 0 dBm (1 mW) and the output signal powers from both the eavesdropper and receiver ports are measured at intervals of 20 s for 5 times by optical power meter (Thorlabs, PM100D). Specially, the output signal powers from the receiver port is retested and collected three times. Depending on the contrast data of the output signal powers between both eavesdropper and receiver ports, it could be found that average 0.13 dB power difference close to 5.84% power splitting ratio is obtained with 30, 50, 70, and 100 mW pumping light power. The detecting accuracy could reach 95.2%. It could be alarmed that there might be the eavesdropper by this method. When the intruder is discovered, some pseudo-code information will be used to confuse eavesdropper by the bottom TCBzC/SU-8 waveguide layer while the top TCNzC/SU-8 waveguide layer is utilized to transfer true information data. 655 nm wavelength signal light pumped by 532 nm wavelength modulating light in the top TCNzC/SU-8 waveguide layer will be transferred into new fiber communication path. The variation of optical operating frequency for both signal and pumping light could also be helpful to ensure the network information security and avoid eavesdropping.

Figure 8. (a) The scheme to detect eavesdropping for the waveguide chip; (b) the output signal powers from the receiver port pumped externally by 30, 50, 70, and 100 mW at 405 nm wavelength, respectively.

The detailed explanation is added into the revised manuscript as follows:

“In our experiment, the approach to monitor the optical transmission loss change in the optical fiber system is adopted to detect the eavesdropping.^[40] The power of the optical signal is a typical and significant parameter, so it could reflect the signal loss by eavesdropping. In **Figure 8a**, the scheme to detect eavesdropping for our proposed optical encryption waveguide chip is described. The optical fiber coupler with 95/5 power splitting ratio is used to demonstrate detect eavesdropping case. 5% signal optical power as eavesdropping influence is measured by the eavesdropper port and 95% signal optical power normally transmitted is tested by the receiver port. To detect the eavesdropping of an intruder, as given in **Figure 8b**, the bottom TCBzC/SU-8 waveguide inputted with 532 nm signal wavelength is pumped

externally by different light powers as 30, 50, 70, and 100 mW at 405 nm wavelength, respectively. The input signal power is set at 0 dBm (1mW) and the output signal powers from both the eavesdropper and receiver ports are measured at intervals of 20 s for 5 times by optical power meter (Thorlabs, PM100D). Specially, the output signal powers from the receiver port is retested and collected three times. Depending on the contrast data of the output signal powers between both eavesdropper and receiver ports, it could be found that average 0.13 dB power difference close to 5.84% power splitting ratio is obtained with 30, 50, 70, and 100 mW pumping light power. The detecting accuracy could reach 95.2%. It could be alarmed that there might be the eavesdropper by this method. When the intruder is discovered, some pseudo-code information will be used to confuse eavesdropper by the bottom TCBzC/SU-8 waveguide layer while the top TCNzC/SU-8 waveguide layer is utilized to transfer true information data. 655 nm wavelength signal light pumped by 532 nm wavelength modulating light in the top TCNzC/SU-8 waveguide layer will be transferred into new fiber communication path. The variation of optical operating frequency for both signal and pumping light could also be helpful to ensure the network information security and avoid eavesdropping.”

[40] Shaneman, K. et al. Optical network security: technical analysis of fiber tapping mechanisms and methods for detection & prevention. *IEEE Military Communications Conference*, Monterey, Canada, 711-716 (2004).

Reviewer's comments (blue) and our answers (black)

Dear Reviewer 2:

We would like to express our gratitude to you for helping us to improve the manuscript. Gratefully thank you for the valuable comments. The main revisions as well as the explanations are shown in the following:

Reviewer #2: In this manuscript, the authors report a dual-layer optical encryption fluorescent polymer waveguide chip based on optical pulse-code modulation technique. The theoretical background and characterizations are relatively comprehensive. This work is interesting, and the manuscript may be recommended for publication after following issues have been addressed successfully.

1. For Figure 2a, the vertical coordinate seemed to be normalized for PL intensity. The author should carefully check this point. In addition, the author should provide the absorbance spectra of TCBzC and TCNzC. I am wondering that whether is there a fluorescence energy resonance transfer (FERT) occurred between TCBzC and TCNzC, which also might their resulting waveguide performance.

Thanks for your comments. According to the reviewer's suggestion, the absorbance spectra of TCBzC and TCNzC are referenced into the manuscript as Ref. [37]. The vertical coordinate of normalized PL intensity has been checked and adjusted according to your suggestion. Based on the spectra, there should be a fluorescence energy resonance transfer (FERT) between TCBzC and TCNzC. Whereas, in our work, the dual-layer optical encryption fluorescent polymer waveguide structure is designed and fabricated. TCBzC and TCNzC are solely used as gain medium for top and bottom waveguide layer, respectively. The P(MMA-co-GMA) buffer layer between the top (TCNzC/SU-8) and bottom (TCBzC/SU-8) waveguide layers could effectively avoid the FERT phenomenon and guarantee the performances of the waveguide chip. The detailed explanation is supplemented into the revised manuscript as follows:

“The normalized PL intensity spectra of TCBzC and TCNzC are depicted in **Figure 2a**. Emission peaks as 540 and 611 nm wavelength are pumped with the 365 nm wavelength light. Absorption peaks as 430 and 505 nm wavelength in visible light region are obtained.^[37] Based on the absorbance spectra, there should be a fluorescence energy resonance transfer (FERT) between TCBzC and TCNzC. In our experiment, the dual-layer optical encryption fluorescent polymer waveguide structure is designed and fabricated. TCBzC and TCNzC are solely used as gain medium for top and bottom waveguide layer, respectively. The P(MMA-co-GMA) buffer layer between the top (TCNzC/SU-8) and bottom (TCBzC/SU-8) waveguide layers could effectively avoid the FERT phenomenon and guarantee the performances of the waveguide chip.”

Figure 2. The normalized PL intensity and absorbance spectra of TCBzC and TCNzC materials.^{[37]}

[37] Wang, C. et al. On-chip optical sources of 3D photonic integration based on active fluorescent polymer waveguide microdisks for light display application. *Photonix* 4, 13 (2023).

2. In this work, why the author choose 655 nm light sources as signal sources while the fluorescence emission of TCNzC is at 610 nm?

Thanks for your comment. The 655 nm wavelength light is a key signal source in visible light fiber communication system.^[38,39] It might not be the maximum fluorescence emission peak of TCNzC (at 610 nm length), but will has greatly potential application in actual optical information transmission network. Therefore, in our experiment, the 655 nm wavelength light is used as signal sources. The description is added into the revised manuscript as follows:

“Depending on the 655 nm wavelength light as key signal source in visible light fiber communication system,^[38,39] it might not be the maximum fluorescence emission peak of TCNzC (at 610 nm length), but will has greatly potential application in actual optical information transmission network.”

[38] Qin, J. et al. Ultrahigh figure-of-merit in metal-insulator-metal magnetoplasmonic sensors using low loss magneto-optical oxide thin films. *ACS Photonics* 4, 1403–1412 (2017).

[39] Reilly, M. A. et al. Optical gain at 650 nm from a polymer waveguide with dye-doped cladding. *Appl. Phys. Lett.* 87, 231116 (2005).

3. As in Figure 7b, what is the main reason for the detectable intermittent bright spots? whether it is caused by light leakage phenomenon, or due to the fluorescence molecules dispersed within polymer matrix? Meanwhile, in Figure 7b, the author should add the bar scale.

Thanks for your comment. The main reason for the detectable intermittent bright spots in **Figure 7b** should be due to the light leakage phenomenon. It might be caused by the actual refractive index perturbation from the thermal stress change between the upper and lower cladding in fabrication process. The bar scale has been added in **Figure 7b** according to your suggestion. These are added into the revised manuscript as follows:

“The main reason for the detectable intermittent bright spots in **Figure 7b** should be due to the light leakage phenomenon. It might be caused by the actual refractive index perturbation from the thermal stress change between the upper and lower cladding in fabrication process.”

Figure 7. The analysis of the optical response properties of the dual-layer waveguide device with 405 nm wavelength pumping light excitation. (a) The top-view microscope when coupling 532 and 655 nm signal lights with 405 nm wavelength light pumping in; (b) the output optical fields of both signal sources ($\times 50$); (c) and (d) are the curves of relative gain versus pumping light power intensity variation for 532 and 655 nm wavelength signal source.

4. The captions of the Figure 8c and 8d are absent.

Thanks for your kind comment. In our revised manuscript, the captions of the **Figure 8c** and **8d** have been supplemented. For the framework of the article, we have combined **Figure 7** and **Figure 8** into new **Figure 7**. The new figure numbers and captions are shown below:

Figure 7. (e) and (f) are the input/output response square-wave with pulse-code modulation for 532 nm wavelength signal source at a frequency of 250 Hz; (g) and (h) are the input/output response square-wave with pulse-code modulation for 655 nm wavelength signal source at a frequency of 250 Hz.

4. In this paper, the corresponding characterization of the optical-loss coefficient of optical waveguide materials should be characterized.

Thanks for your comment. According to your suggestion, the varied curves for refractive index (n) and extinction coefficient (k) in Vis-NIR wavelength region for TCBzC/SU-8 and TCNzC/SU-8 waveguide materials are measured by the ellipsometer (SPEL M-2000VI, America) as shown in **Figure 1c**. Based on the complex refractive index (\tilde{n}) function of $\tilde{n} = n + ik$, k refers to the imaginary part of the optical constant as optical absorption-loss coefficient. Therefore, the value of k could be defined as the main optical-loss coefficient for TCBzC/SU-8 and TCNzC/SU-8 waveguide materials. In **Figure 1c**, it could be found that the values of k for both TCBzC/SU-8 waveguide material at 532 nm wavelength and TCNzC/SU-8 waveguide material at 655 nm wavelength are less than $3 \times 10^{-4} \text{ cm}^{-1}$. It is demonstrated that TCBzC/SU-8 waveguide material at 532 nm wavelength and TCNzC/SU-8 waveguide material at 655 nm wavelength have low optical-loss characterization. These are added into the revised manuscript as follows:

“The varied curves for refractive index (n) and extinction coefficient (k) in Vis-NIR wavelength region for TCBzC/SU-8 and TCNzC/SU-8 waveguide materials are measured by the ellipsometer as **Figure 1c**. According to the complex refractive index (\tilde{n}) function of $\tilde{n} = n + ik$, k refers to the imaginary part of the optical constant as optical absorption-loss coefficient. Therefore, the value of k could be defined as the main optical-loss coefficient for TCBzC/SU-8 and TCNzC/SU-8 waveguide materials. In **Figure 1c**, it could be found that the values of k for both TCBzC/SU-8 waveguide material at 532 nm wavelength and TCNzC/SU-8 waveguide material at 655 nm wavelength are less than $3 \times 10^{-4} \text{ cm}^{-1}$. It is demonstrated that TCBzC/SU-8 waveguide material at 532 nm wavelength and TCNzC/SU-8 waveguide material at 655 nm wavelength have low optical-loss characterization.”

Figure 1. (c) The refractive index (n) and extinction coefficient (k) characterization of TCBzC/SU-8 and TCNzC/SU-8 optical waveguide materials.

We have made careful modifications to the reviewers’ questions. Our work has been greatly improved according to the reviewers’ comments and we hope that the revised manuscript can meet the requirements of reviewers.

Reviewer #1 (Remarks to the Author):

The authors have satisfactorily addressed all my concerns and questions. I am pleased to recommend the manuscript for publication.

Reviewer #2 (Remarks to the Author):

The author has responded well to the questions raise. This manuscript can be accepted for publication

Reviewer's comments (blue) and our answers (black)

Reviewer #1:

The authors have satisfactorily addressed all my concerns and questions. I am pleased to recommend the manuscript for publication.

We would like to express our gratitude to your help for improving the manuscript.

Reviewer #2:

The author has responded well to the questions raise. This manuscript can be accepted for publication.

We would like to express our gratitude to your help for improving the manuscript.